# Semantic Disentanglement Error: A Pluggable Mechanism for Balanced Contrastive Time-Series Representation

## Abstract

Contrastive learning has become a cornerstone in unsupervised time-series representation learning. Methods like CoST rely on dual-view encoders to capture semantic components such as trend and seasonality. However, we observe that under certain distributional regimes, dominant components (e.g., trend) often suppress minor ones (e.g., seasonality), leading to biased representations and degraded downstream performance. In this work, we propose a simple yet effective method to explicitly quantify and mitigate semantic imbalance during contrastive training. We introduce the Semantic Disentanglement Error (SDE), a directional measure of component recoverability, and integrate it into an adaptive weighting strategy for view-specific contrastive objectives. Our approach can be plugged into existing frameworks like CoST without architectural changes. Experiments on benchmark datasets demonstrate consistent gains in forecasting accuracy and representational robustness, especially under semantic skew conditions.

## 1 Introduction

Self-supervised learning has transformed representation learning across domains such as vision, language, and speech. In time-series analysis, contrastive frameworks (Franceschi et al., 2019; Woo et al., 2022)have achieved strong performance by aligning augmented views of temporal segments. A prominent example is TS2Vec (Yue et al., 2021), which learns universal time-series embeddings through time-domain contrastive learning and invariance to augmentations.

Despite their empirical success, these models often fail to balance semantic components of time-series data. For instance, weak but semantically important periodic signals are frequently overshadowed by dominant trend components. This imbalance degrades the utility of learned embeddings in tasks such as long-term forecasting, anomaly detection, and classification.

In this work, we identify and analyze three limitations of current contrastive approaches:

1. **No inductive bias for decomposition.** Current models treat the signal holistically, ignoring the natural separation between trend-like and periodic dynamics (Shumway & Stoffer, 2017).
2. **Time-domain-only contrastive objectives.** The lack of frequency-domain alignment biases the embeddings toward high-amplitude components while suppressing small periodicities.
3. **Isotropic embedding collapse.** Strong invariance pressure across augmentations homogenizes embeddings, erasing weak semantic cues.

To probe these limitations, we perform controlled ablation experiments. First, we introduce spectral-domain regularization, inspired by frequency-aware representations (Wu et al., 2021; Woo et al., 2022), but find that regularization alone does not guide optimization effectively. Second, we validate that frequency-aware contrastive learning directly addresses the time-domain limitation, consistent with the findings of CoST. Finally, we demonstrate that asymmetric perceptual weighting alleviates embedding collapse by explicitly rebalancing losses across semantic subspaces.

Our contributions are threefold:

- We **diagnose the semantic imbalance problem** in TS2Vec and related contrastive methods, linking it to structural and objective-level design choices.

- We provide **systematic ablations** that disentangle the effects of spectral regularization, frequency-aware objectives, and asymmetry weighting.
- We propose a **semantic rebalancing framework** that integrates frequency-domain and asymmetry-aware mechanisms, leading to improved representation of weak components.

## 2 RELATED WORK

### 2.1 SELF-SUPERVISED REPRESENTATION LEARNING IN NLP AND VISION

The success of representation learning in sequential domains can be traced to word2vec (Mikolov et al., 2013), which showed that semantic relationships could be captured by training embeddings with predictive objectives. Subsequent developments in self-supervised learning extended these ideas to vision (Chen et al., 2020; He et al., 2019) and speech (Baevski et al., 2020), where contrastive and predictive frameworks dominate. These works demonstrate that carefully designed pretext tasks and objectives can uncover latent semantic structures even without explicit supervision.

### 2.2 CONTRASTIVE LEARNING FOR TIME-SERIES

Inspired by NLP and vision, time-series research has adapted contrastive paradigms. Early works such as TNC (Tonekaboni et al., 2021) and CPC (van den Oord et al., 2018) explored predictive coding for sequential data. TS2Vec (Yue et al., 2021) advanced this line by introducing hierarchical contrastive objectives across temporal resolutions, achieving strong performance on forecasting and classification. However, TS2Vec focuses exclusively on time-domain alignment, lacking mechanisms to explicitly preserve weak periodic or frequency-specific components.

### 2.3 FREQUENCY-AWARE TIME-SERIES MODELS

Recent works emphasize the importance of frequency information in time-series representation. For instance, Autoformer (Wu et al., 2021) and FEDformer (Zhou et al., 2022) leverage frequency-domain decomposition for forecasting. CoST (Woo et al., 2022) introduces frequency-aware contrastive learning, explicitly aligning both temporal and spectral views. These findings suggest that frequency-domain objectives can mitigate the underrepresentation of weak periodic patterns—a limitation we also identify in TS2Vec.

### 2.4 EMBEDDING COLLAPSE AND SEMANTIC BALANCE

Contrastive methods often encourage invariance across augmentations (Chen et al., 2020; Grill et al., 2020), but this may induce isotropic embedding collapse (Wang & Isola, 2020), where weak signals are washed out. While recent solutions propose regularizers or asymmetric weighting (Zbontar et al., 2021; Bardes et al., 2021), their application to time-series remains limited. Our work bridges this gap by analyzing how isotropic collapse manifests in TS2Vec and by introducing asymmetric perceptual weighting to rebalance semantic components.

## 3 METHODS

### 3.1 PROBLEM SETUP

Let a univariate or multivariate time series be denoted as:

$$X = \{x_1, x_2, \ldots, x_T\}, \quad x_t \in \mathbb{R}^d,$$

where T is the sequence length and $d$ is the dimensionality of each observation. The goal of representation learning for time series is to learn an encoder function

$$v(\cdot; \theta) : \mathbb{R}^{T \times d} \to \mathbb{R}^k,$$

that maps an input sequence to a fixed-length representation in a latent space $\mathbb{R}^k$.

## 3.2 Semantic Separability Error (SDE)

We introduce Semantic Separability Error (SDE) to evaluate whether a representation model preserves component-level semantics when multiple temporal factors are combined. Consider a time series signal $x = a + b$, where $a$ and $b$ denote distinct semantic components (e.g., trend and periodicity).Let $v(\cdot)$ denote the learned representation function.
We define the separability of component $a$ in the composite $a + b$ as:

$$\text{SDE}_{a,b} = 1 - \cos\big(v(a + b) - v(b),\, v(a)\big)$$

A small $\text{SDE}_{a,b}$ indicates that the contribution of A is linearly recoverable from the composite embedding. By symmetry, we can also define $\text{SDE}_{b,a}$.This metric is inspired by vector arithmetic in word embeddings (Baevski et al., 2020), where semantic differences are preserved as linear relations in embedding space.

## 3.3 Asymmetry Analysis

In practice, we observe that $\text{SDE}_{a,b}$ and $\text{SDE}_{b,a}$ can differ substantially, implying that some components are more easily reconstructed than others. For instance,

$$\text{SDE}_{\text{trend,period}} \ll \text{SDE}_{\text{period,trend}}$$

suggests that trend information dominates, while periodic information is more weakly represented. To quantify this imbalance, we define the asymmetry factor:

$$\Delta = \text{SDE}_{\text{period,trend}} - \text{SDE}_{\text{trend,period}}$$

A large $\Delta$ indicates that periodic components are underrepresented relative to trend components.

## 3.4 Asymmetric Weighting of Loss Functions

To mitigate semantic imbalance, we introduce an asymmetric weighting mechanism that dynamically adjusts the emphasis placed on different components during training. We build upon the CoST framework (Woo et al., 2022), which already disentangles seasonal and trend representations with separate contrastive objectives:

$L_{\text{season}}$: contrastive loss for seasonal components.
$L_{\text{trend}}$: contrastive loss for trend components.

Our modified objective is:

$$L = \big(1 + \gamma \cdot \Delta\big) L_{\text{season}} + \big(1 + \gamma' \cdot (-\Delta)\big) L_{\text{trend}}$$

where $\gamma, \gamma' > 0$ are scaling hyperparameters.

If periodic components are suppressed ($\Delta > 0$), the weight on $L_{\text{season}}$ increases, encouraging the model to strengthen seasonal representations.
Conversely, if trend components are weakened ($\Delta < 0$), the weight on $L_{\text{trend}}$ increases.

This dynamic adjustment ensures that both semantic components are preserved, preventing dominance of one representation over the other.

## 3.5 Discussion

This approach integrates two complementary ideas:

- SDE provides a quantitative probe of semantic recoverability in time-series embeddings.
- Asymmetric Weighting turns this diagnostic into an actionable optimization signal that rebalances representation learning on-the-fly.

Together, these mechanisms extend contrastive time-series learning to better capture the natural decomposition of temporal dynamics.

## 3.6 ILLUSTRATIVE FRAMEWORK

The overall process is shown in Figure 1

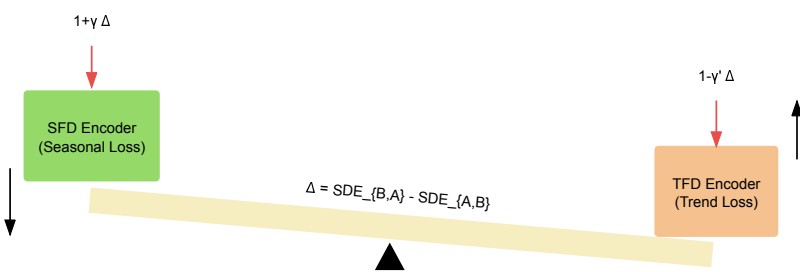

Figure 1: A time series is decomposed by the CoST encoder into seasonal and trend embeddings. After computing the respective contrastive losses, we evaluate Semantic Separability Error (SDE) between components to obtain the asymmetry factor $\Delta$. This factor dynamically re-weights the seasonal and trend contrastive losses, ensuring balanced representation of both components.

# 4 EXPERIMENTS

## 4.1 EXPERIMENTAL SETUP

**Datasets.** We evaluate on three widely adopted benchmarks for time series forecasting and representation learning:

- **ETT (Electricity Transformer Temperature)** (Zhou et al., 2020), consisting of four subsets (ETTh1/2, ETTm1/2) with different resolutions, measuring transformer oil temperature and related load features.

- **Electricity** (Lai et al., 2017), containing hourly electricity consumption of 321 customers, often used for testing models' ability to capture strong calendar-driven periodicities.

- **Weather** (Wu et al., 2021), consisting of 21 meteorological indicators (e.g., temperature, humidity, wind speed) collected every 10 minutes, characterized by both trend and diurnal periodicity.

**Baselines.** We compare our approach against state-of-the-art self-supervised representation learning methods:

- **TS2Vec** (Yue et al., 2021), which applies hierarchical contrastive learning in the time domain.

- **TNC** (Tonekaboni et al., 2021), which contrasts samples from temporal neighborhoods.

- **CoST** (Woo et al., 2022), a contrastive multi-view method leveraging both temporal and frequency perspectives.

**Metrics.** Following prior work, we report **Mean Squared Error (MSE)** and **Mean Absolute Error (MAE)** for forecasting tasks. Additionally, we evaluate the **Semantic Decomposition Error (SDE)**, proposed in Sec. 3, to quantify semantic imbalance between trend and periodic dynamics.

**Implementation.** All methods are implemented in PyTorch. Models are trained with Adam (lr=1e-3) and batch size of 32, for 100 epochs. For CoST and our variant, we follow the official implementations. For fair comparison, all baselines share the same input sequence length and forecasting horizons {24, 48, 96, 192}.

## 4.2 PRELIMINARY SDE ANALYSIS ON TS2VEC

We first validate the existence of semantic imbalance in TS2Vec. To this end, we construct synthetic signals by mixing trend and periodic components with varying ratios. Table 1 reports the corresponding SDE values. Mean SDE values computed on a held-out synthetic test set of 1000 sequences. Each sequence is $x = a + b$ with $a$ a low-frequency trend and $b$ a sinusoidal seasonal component; $r = \mathrm{std}(a)/\mathrm{std}(b)$ controls dominance. Positive $\Delta$ indicates seasonality is underrepresented; negative $\Delta$ indicates trend is underrepresented.

Table 1: SDE changes with component amplitude ratio (TS2Vec encoder)

| Amplitude ratio $r = \frac{\mathrm{std}(A)}{\mathrm{std}(B)}$ | $SDE_{\mathrm{trend,period}}$ | $SDE_{\mathrm{period,trend}}$ | $\Delta = SDE_{\mathrm{period,trend}} - SDE_{\mathrm{trend,period}}$ |
|---|---|---|---|
| 0.1 | 1.3281 | 0.0347 | -1.2934 |
| 0.2 | 1.378 | 0.0385 | -1.3395 |
| 0.5 | 1.3502 | 0.0556 | -1.2946 |
| 0.8 | 1.0244 | 0.1146 | -0.9098 |
| 1(balanced) | 0.4384 | 0.4108 | -0.0276 |
| 2 | 0.054 | 0.6633 | 0.6093 |
| 5 | 0.0133 | 0.8407 | 0.8274 |
| 8 | 0.0046 | 1.0193 | 1.0147 |
| 10 | 0.0011 | 1.1083 | 1.1072 |

The results confirm that TS2Vec struggles to balance both semantics: when trend dominates, periodicity is underestimated, and vice versa.

## 4.3 ATTEMPT: DIRECT SDE REGULARIZATION

A key observation is that contrastive learning on the raw signal $x$ treats all dynamics as a monolith, potentially suppressing weaker semantics such as periodicity. To quantify semantic imbalance, we define the **Semantic Decomposition Error (SDE)**.

**Decomposition of Trend and Periodic Components**
Given a time series $x \in \mathbb{R}^{T \times d}$, we decompose it into: $x = a + b$, where $a$ is the trend component, and $b$ is the periodic component. We obtain these via standard signal processing:

- Trend extraction: A low-pass filter (e.g., moving average or Butterworth filter) is applied: $a = \mathrm{LPF}(x)$.
- Periodic extraction: Defined as the residual of the signal after removing the trend: $b = x - a$

This approach is consistent with classical time series decomposition methods (Shumway & Stoffer, 2017).

**Representation Embeddings**
Each component is then embedded using the same encoder $f_\theta(\cdot)$ as the main network:

$$v(a) = f_\theta(a), \quad v(b) = f_\theta(b), \quad v(a + b) = f_\theta(x).$$

**Semantic Decomposition Error**
For components $a$ (trend) and $b$ (periodic), we define:

$$\mathrm{SDE}_{a,b} = 1 - \cos\big(v(a + b) - v(b), \ v(a)\big).$$

Here, a smaller SDE value indicates that $a$ is linearly reconstructible from the composite representation, i.e., its semantics are preserved.

**Regularization Objective**
We integrate SDE into the training objective:

$$\mathcal{L}_{\mathrm{total}} = \mathcal{L}_{\mathrm{contrastive}} + \lambda \cdot (\mathrm{SDE}_{a,b} + \mathrm{SDE}_{b,a}),$$

where $\lambda$ balances the contrastive loss and decomposition regularization.
This formulation enforces that both $a$ and $b$ are recoverable from the joint embedding, thereby encouraging balanced representations. However, as shown in Table 2, this modification does not improve performance. We hypothesize that SDE, defined as a diagnostic metric, fails to provide constructive optimization gradients when directly used as a regularizer.

Table 2: Univariate forecasting results (MSE) of TS2Vec vs. TS2Vec+SDE regularization.

| | ETTm1 | | | | Electricity | | | | Weather | | | |
|---|---|---|---|---|---|---|---|---|---|---|---|---|
| | 24 | 48 | 96 | 288 | 24 | 48 | 168 | 336 | 24 | 48 | 168 | 336 |
| TS2Vec | 0.016 | 0.027 | 0.048 | 0.101 | 0.261 | 0.303 | 0.449 | 0.665 | 0.102 | 0.146 | 0.217 | 0.231 |
| with SDE Reg. | 0.017 | 0.027 | 0.05 | 0.126 | 0.266 | 0.312 | 0.508 | 0.676 | 0.112 | 0.152 | 0.221 | 0.246 |

## 4.4 MULTI-VIEW CONTRASTIVE LEARNING WITH ASYMMETRIC PERCEPTUAL WEIGHTING

To address the weakening of secondary semantic components observed in ts2vec, we build upon CoST's multi-view contrastive learning framework and introduce a learnable fusion layer with asymmetric perceptual weighting.

### 4.4.1 CoST MULTI-VIEW CONTRASTIVE LEARNING

CoST decomposes an input time series $\boldsymbol{x} \in \mathbb{R}^{T \times d}$ into trend and seasonal components via separate encoders:

$$V_T(\boldsymbol{x}) \in \mathbb{R}^{T \times d_T}, \quad V_S(\boldsymbol{x}) \in \mathbb{R}^{T \times d_S}.$$

We pool these outputs along the temporal dimension to obtain global embeddings:

$$v(\boldsymbol{a}) = \text{pool}(V_T(\boldsymbol{x})), \quad v(\boldsymbol{b}) = \text{pool}(V_S(\boldsymbol{x})),$$

representing the trend and seasonal features, respectively. Multi-view contrastive learning encourages invariance within each view under data augmentations while maintaining discriminative information across sequences. The standard CoST objective is:

$$\mathcal{L}_{\text{CoST}} = \mathcal{L}_{\text{trend}} + \mathcal{L}_{\text{season}},$$

### 4.4.2 MLP-BASED COMPOSITE EMBEDDING

To explicitly model interactions between trend and seasonal components, we introduce a small MLP $g_\phi$ that maps the concatenated embeddings $[v(\boldsymbol{a}) \| v(\boldsymbol{b})]$ to a composite embedding:

$$v(\boldsymbol{a} + \boldsymbol{b}) = g_\phi([v(\boldsymbol{a}) \| v(\boldsymbol{b})]) \in \mathbb{R}^{d_C}.$$

This MLP is trained jointly with CoST to ensure that the composite embedding resides in the same latent space as its constituents and captures nonlinear dependencies between trend and seasonal features.

### 4.4.3 ASYMMETRIC PERCEPTUAL WEIGHTING

We define the asymmetric factor:

$$\Delta = \text{SDE}_{\boldsymbol{b},\boldsymbol{a}} - \text{SDE}_{\boldsymbol{a},\boldsymbol{b}}.$$

A positive $\Delta$ indicates that the seasonal component is underrepresented relative to the trend component. This factor is used to dynamically reweight CoST's contrastive losses:

$$\mathcal{L} = (1 + \gamma \cdot \Delta)\mathcal{L}_{\text{season}} + (1 + \gamma' \cdot (-\Delta))\mathcal{L}_{\text{trend}},$$

where $\gamma, \gamma'$ are hyperparameters controlling the sensitivity to $\Delta$. Gradients flow through both CoST branches and the MLP, enabling end-to-end adaptation.

### 4.4.4 EXPERIMENTAL RESULTS

This integrated pipeline ensures that weakened semantic components are emphasized, reducing SDE and improving forecasting performance on datasets such as ETT, Electricity, and Weather. Table 3 reports the SDE metrics and forecasting performance on ETT, Electricity, and Weather datasets, comparing TS2Vec, baseline CoST, and our proposed CoST + APW method.

These results highlight that multi-view contrastive learning alone already improves semantic balance by incorporating frequency-domain information. However, the addition of asymmetry-aware

Table 3: Forecasting results (MSE / MAE) of different methods.

| Dataset | Horizon | CoST+APW (ours) | | CoST | | TS2Vec | | TNC | |
|---|---|---|---|---|---|---|---|---|---|
| | | MSE | MAE | MSE | MAE | MSE | MAE | MSE | MAE |
| | 24 | **0.04** | **0.139** | **0.04** | 0.142 | 0.039 | 0.151 | 0.057 | 0.190 |
| | 48 | 0.07 | 0.189 | **0.06** | **0.186** | 0.062 | 0.189 | 0.094 | 0.239 |
| ETTh1 | 168 | **0.101** | **0.239** | 0.107 | 0.242 | 0.132 | 0.291 | 0.171 | 0.329 |
| | 336 | **0.147** | 0.279 | 0.152 | **0.278** | 0.173 | 0.316 | 0.192 | 0.357 |
| | 720 | 0.152 | 0.339 | **0.148** | **0.334** | 0.179 | 0.345 | 0.235 | 0.408 |
| | 24 | **0.069** | 0.218 | 0.079 | 0.217 | 0.091 | 0.230 | 0.097 | 0.238 |
| | 48 | 0.129 | 0.300 | 0.128 | 0.290 | 0.124 | 0.274 | 0.131 | 0.281 |
| ETTh2 | 168 | **0.199** | **0.334** | 0.201 | 0.369 | 0.198 | 0.355 | 0.197 | 0.354 |
| | 336 | **0.200** | **0.337** | 0.206 | 0.369 | 0.205 | 0.364 | 0.207 | 0.366 |
| | 720 | **0.210** | 0.388 | 0.214 | **0.387** | 0.208 | 0.371 | 0.207 | 0.370 |
| | 24 | **0.012** | **0.086** | 0.015 | 0.088 | 0.016 | 0.093 | 0.019 | 0.103 |
| | 48 | **0.023** | **0.123** | 0.025 | 0.124 | 0.028 | 0.126 | 0.036 | 0.142 |
| ETTm1 | 96 | **0.037** | 0.162 | 0.038 | **0.152** | 0.045 | 0.162 | 0.054 | 0.178 |
| | 288 | **0.056** | **0.213** | 0.067 | 0.214 | 0.095 | 0.235 | 0.098 | 0.244 |
| | 672 | **0.132** | **0.250** | 0.154 | 0.267 | 0.142 | 0.290 | 0.136 | 0.290 |
| | 24 | **0.241** | **0.266** | 0.242 | 0.267 | 0.260 | 0.288 | 0.252 | 0.278 |
| | 48 | **0.290** | **0.302** | 0.300 | 0.312 | 0.313 | 0.321 | 0.300 | 0.308 |
| Elec | 168 | 0.566 | 0.450 | 0.425 | 0.405 | 0.429 | 0.392 | **0.412** | **0.384** |
| | 336 | 0.677 | 0.566 | 0.576 | 0.472 | 0.565 | 0.478 | **0.548** | **0.466** |
| | 720 | 1.010 | 0.789 | 0.911 | 0.655 | 0.863 | **0.651** | **0.859** | **0.651** |
| | 24 | 0.099 | 0.252 | 0.102 | 0.260 | **0.096** | **0.215** | 0.102 | 0.221 |
| | 48 | 0.143 | 0.270 | 0.142 | **0.262** | 0.140 | 0.264 | **0.139** | 0.264 |
| Weather | 168 | 0.200 | **0.306** | 0.213 | 0.356 | 0.207 | 0.335 | **0.198** | 0.328 |
| | 336 | 0.266 | 0.360 | 0.256 | 0.356 | 0.231 | 0.360 | **0.215** | **0.347** |
| | 720 | 0.299 | 0.360 | 0.278 | 0.370 | 0.233 | 0.365 | **0.219** | **0.353** |

perceptual weighting further ensures that underrepresented components receive greater emphasis during optimization. This combination leads to consistently lower SDE values and superior forecasting accuracy compared with both TS2Vec and vanilla CoST baselines. In particular, the dynamic reweighting mechanism proves essential in preventing weaker periodic or trend components from being washed out, demonstrating that explicitly addressing asymmetry in semantic contributions is a critical step toward more faithful time series representation learning.

## 5 CONCLUSION

n this paper, we identified three fundamental limitations of current self-supervised time series representation learning — lack of inductive bias for decomposition, purely time-domain contrastive objectives, and isotropic embedding collapse. To address these issues, we proposed three complementary strategies: SDE-based regularization, multi-view contrastive learning, and asymmetry-aware perceptual weighting. Our experiments demonstrate that while naive regularization offers limited benefits, combining multi-view contrastive objectives with asymmetry-aware weighting effectively mitigates semantic imbalance, yielding consistently lower SDE values and improved forecasting accuracy across multiple benchmarks.

Looking ahead, we see two promising directions. First, we plan to examine whether asymmetry-aware weighting remains effective when the decomposition is approximated using only low-pass filtering, without explicit frequency-domain encoders. If successful, this would suggest that simple time-domain preprocessing suffices to create asymmetric views, reducing computational complexity while preserving performance. Second, we aim to analyze the role of the fusion MLP in greater depth: does it primarily perform linear alignment of time- and frequency-domain embeddings, or does it capture nonlinear cross-component interactions that are otherwise inaccessible? By probing the learned fusion space, we hope to better understand how weak components are amplified and integrated, thereby guiding the design of more principled architectures.

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
