# OpenReview forum: "Semantic Disentanglement Error: A Pluggable Mechanism for Balanced Contrastive Time-Series Representation"
_ICLR.cc/2026/Conference — Submitted to ICLR 2026_

### Official Review · Reviewer_paab · 2025-10-25

**Soundness:** 2
**Presentation:** 2
**Contribution:** 1
**Rating:** 0
**Confidence:** 5

**Summary:**

This paper proposes Semantic Disentanglement Error (SDE), a quantitative metric to diagnose and mitigate semantic imbalance in contrastive time-series representation learning. The method introduces SDE as a direction-aware measure between decomposed trend and seasonal components, and further integrates it into an asymmetry-aware adaptive weighting scheme that dynamically rebalances contrastive losses across these components.

**Strengths:**

The proposed SDE provides an interpretable metric that directly links representation geometry to semantic disentanglement, and the asymmetry-aware adaptive weighting effectively rebalances learning signals.

**Weaknesses:**

1. The article is too short, with less than 7 pages of main text and 8 pages of references. The content is not comprehensive and is not conducive to understanding

2. While SDE is conceptually intuitive, the paper lacks formal analysis on why asymmetric weighting ensures balanced gradients or avoids representation collapse. The directional term ($\Delta = \mathrm{SDE}_{B,A} - \mathrm{SDE}_{A,B}$) is justified empirically, but no convergence or stability theorem is provided. A geometric or optimization-theoretic explanation of how SDE relates to disentanglement would strengthen the contribution.

3. There are too few baseline models for comparison. Please add more baselines published in the past 2 years for comparison.

4. The ablations confirm SDE’s contribution but do not visualize how semantic balance changes representation geometry. Embedding plots (e.g., t-SNE/UMAP) or reconstructed component visualization would better substantiate the “semantic disentanglement” claim.

5. Table 1 format is not standardized.

6. Insufficient experiments and lack of experimental analysis.

7. The article still has a lot of page space, but the introduction, related work， The experimental analysis is described briefly, lacking in richness and content.

**Questions:**

Please see the above Weakness and the below:

1. Please offer a more formal justification—e.g., via gradient norm analysis or optimization geometry—that minimizing SDE leads to balanced representation learning between semantic components.

2. Please provide the Hyperparameter Sensitivity, such as γ and γ′, and does the adaptive weighting across datasets with different noise or periodicity.

3. Can you visualize the embeddings or reconstruction results to show how SDE changes the structure of the latent space compared to CoST?

4. Does SDE affect contrastive uniformity or isotropy (e.g., risk of representation collapse)? Have you tested variance or eigenvalue spectra before and after applying SDE?

---

### Official Review · Reviewer_cZ46 · 2025-10-30

**Soundness:** 2
**Presentation:** 2
**Contribution:** 1
**Rating:** 2
**Confidence:** 5

**Summary:**

The paper targets semantic imbalance in contrastive time-series representation learning. It introduces Semantic Disentanglement (Separability) Error (SDE) as a directional, component-recoverability metric and an asymmetry factor $\Delta = \mathrm{SDE}_{B,A} - \mathrm{SDE}_{A,B}$ to quantify bias between components. Building on CoST, the authors propose a pluggable asymmetric perceptual weighting (APW) that dynamically re-weights seasonal vs. trend contrastive losses using $\Delta$.

**Strengths:**

A practical and plug-in mechanism to measure and correct semantic imbalance in TS contrastive learning: SDE/$\Delta$ provides interpretable diagnostics. APW turns them into on-the-fly loss re-weighting that integrates smoothly with CoST.

**Weaknesses:**

1. This paper as a whole seems like a semi-finished product, with insufficient experimental analysis, too few baseline comparisons, and insufficient richness.

2. The content is insufficient, the main text is less than 7 pages, and there are no appendix.

3. The code is also not publicly available and cannot be obtained.

4. No convergence or stability analysis for APW’s $\Delta$-driven weights.

**Questions:**

The above weakness and the following:

1. Beyond intuition, can you show a toy proof that APW reduces an imbalance surrogate under stationary noise?

2. Provide sensitivity of $\gamma, \gamma'$, and whether a learned gate outperforms fixed scalars.

3. Add more experiments to enrich the content.

4. Report wall-time/throughput vs. CoST.

---

### Official Review · Reviewer_Najj · 2025-10-30

**Soundness:** 1
**Presentation:** 1
**Contribution:** 1
**Rating:** 0
**Confidence:** 5

**Summary:**

The paper introduces Semantic Disentanglement Error (SDE) and an asymmetric weighting scheme to address semantic imbalance in contrastive time-series representation learning. The idea is to quantify component recoverability between trend and seasonal parts and use it to reweight losses. Experiments on ETT, Electricity, and Weather show moderate improvements over CoST.

**Strengths:**

The proposed method can be plugged into existing frameworks without structural change.

**Weaknesses:**

The novelty is marginal

Improvements over baselines are small and sometimes inconsistent.

The analysis lacks theoretical grounding; why SDE provides meaningful gradients is not convincingly explained.

Experiments focus on limited datasets and do not evaluate robustness or scalability.

**Questions:**

See weakness

---

### Official Review · Reviewer_SkhB · 2025-10-31

**Soundness:** 1
**Presentation:** 1
**Contribution:** 1
**Rating:** 0
**Confidence:** 4

**Summary:**

This paper addresses the problem of semantic imbalance in unsupervised time-series representation learning. The authors find that existing models often fail to adequately capture minor semantic components (e.g., seasonality) as they are typically suppressed by dominant ones (e.g., trend). To mitigate this, the authors propose SDE as a quantitative metric and design an adaptive weighting mechanism that incorporates SDE into the contrastive loss function. The problem addressed in the paper is important, but the novelty is somewhat limited and the paper is not presented clearly enough.

**Strengths:**

1. The experimental results provided show that the proposed method has improvement.

**Weaknesses:**

1. The novelty of this work is modest. It introduces the SDE metric but fails to provide sufficient evidence to validate its correctness.Moreover, the integration of SDE into the loss function via a basic weighting mechanism is rather simple, and the paper lacks a thorough analysis of its working mechanism.
2. The scope of the related work and baselines is outdated. Is the field truly devoid of recent advances?
3. The paper's structure could be improved. The methodological description is repeated between Section 3 and Section 4.4, creating redundancy and confusion for the reader.
4. The manuscript contains several basic errors that undermine its precision. For instance, key equations lack numbering, and undefined symbols (e.g., 'T' on line 103, 'A' on line 117) appear in the text without corresponding definitions in the formulas.

**Questions:**

1. Basic errors and structural issues must be rectified.
2. The scope of related work and baselines must be expanded to include recent state-of-the-art references.
3. The analysis of the proposed SDE and adaptive weighting mechanism requires a more thorough and insightful treatment.

---

### Meta-Review · Area_Chair_Mx3s · 2026-01-02

**Summary:**

Reviewers agree that while the paper addresses a relevant problem and reports some performance improvements, its overall contribution is limited. The proposed SDE metric and adaptive weighting mechanism lack sufficient theoretical justification and are technically simple, offering only marginal novelty over existing approaches. Experimental gains are modest and evaluated on a narrow set of datasets with outdated baselines. In addition, the manuscript suffers from significant presentation and completeness issues, including insufficient analysis, missing definitions, and structural redundancy. These concerns are fundamental rather than incremental, leading to a clear recommendation to reject.

**Reviewer Concerns:**

The authors did not provide a rebuttal. As a result, none of the reviewers’ concerns were directly addressed.

**Reviewer Scores:**

As no rebuttal was provided, there was no new information to prompt score changes. Given the reviewers’ high confidence and fundamental concerns, the scores would likely remain unchanged.

---

### Decision · Program_Chairs · 2026-01-26

Reject